# Investigation on Thermal Barrier Coating Damages for Ultrafast Laser Drilling of Cooling Holes

**DOI:** 10.3390/ma16155395

**Published:** 2023-07-31

**Authors:** Shaojian Wang, Yue Hu, Yangjie Zuo, Zenan Yang, Ruisong Jiang

**Affiliations:** 1School of Aeronautics and Astronautics, Sichuan University, Chengdu 610200, China; 2022226210010@stu.scu.edu.cn (S.W.); jushushu25@163.com (Y.H.); zuoyangjie@scu.edu.cn (Y.Z.); 2Science and Technology on Advanced High Temperature Structural Materials Laboratory, Beijing Institute of Aeronautical Materials, Beijing 100095, China; yangzn719@126.com

**Keywords:** picosecond laser, thermal barrier coating microholes, processing quality

## Abstract

To mitigate the challenges pertaining to coating damage and processing defects arising from the utilization of ultrafast laser drilling for microhole creation in thermal barrier coatings (TBCs), thereby exerting substantial influence on the long-term durability of these microholes, the investigation proposes a comprehensive methodology. It encompasses the design of a two-factor four-level full factorial experiment and the execution of experimental research on picosecond laser drilling of TBC microholes. By meticulously analyzing the morphology of the microholes and the coating interface, the damage mechanisms associated with picosecond laser drilling of TBC microholes, as well as the influence of laser process parameters on coating damage, are revealed. The findings reveal that the optimal microhole entrance quality and the lowest roughness along the hole perimeter are attained at a laser power of 12 W and a scanning speed of 320 mm/s. Moreover, at a laser power of 30 W and a scanning speed of 320 mm/s, the minimal crack length on the blunt angle side of the hole and the highest machining quality are observed.

## 1. Introduction

The aviation engine is a vital constituent of an aircraft, often hailed as its “heart”. To further enhance the thrust-to-weight ratio of aviation engines, it is imperative to elevate the turbine inlet temperature, imposing higher demands on the heat tolerance of turbine blades. Currently, turbine blades widely employ thermal barrier coatings (TBCs) and incorporate composite film cooling techniques, utilizing single-crystal hollow blades. These advancements significantly bolster the blades’ capacity to withstand elevated temperatures [1,2], as illustrated in Figure 1.

The conventional processes of electrical discharge machining (EDM) and electrochemical jet machining are unable to fabricate film cooling holes with thermal barrier coatings (TBCs). As a result, the coating process must be performed after the drilling, which can potentially lead to blockage of the film cooling holes [3]. Ultrafast laser machining has emerged as an ideal method for achieving precise manufacturing of film cooling holes on turbine blades with thermal barrier coatings, thanks to its non-selectivity to materials, absence of mechanical strain, and ability to cover large areas [4]. However, the inherent non-linear effects, diffraction, and plasma splashing associated with lasers may pose a risk of damaging the thermal barrier coatings. Therefore, further comprehensive research in this area is necessary.

Researchers have conducted studies on the preparation of thermal barrier coating (TBC) microholes using ultrafast laser technology. Wang et al. [5] investigated the influence of self-focusing and waveguide effects on the propagation behavior of laser beams in shallow micro-holes. The results indicated that the number of spiral orbits required for through-hole formation remains relatively stable within a certain defocusing range. Additionally, self-focusing still significantly enhances the intracavity light intensity as the aperture size increases, while the waveguide effect on sidewalls weakens the intensity enhancement. The authors proposed a two-step processing method involving millisecond lasers for initial hole drilling and subsequent improvement using femtosecond lasers, which were experimentally validated for their effectiveness [6].

In a study by Lu et al. [7], microholes were fabricated on 2.3 mm thick nickel-based high-temperature alloys with TBCs using femtosecond lasers. The researchers analyzed the influence of laser focal position and power on the morphology of the holes. The results showed that a negative defocusing of the laser focal position was required during the processing to ensure the highest laser energy position inside the material. They also identified the corresponding defocusing distances for different laser energies. Furthermore, it was found that high-quality microholes could be achieved using low-power femtosecond lasers (1.5 W) with relatively low technical requirements. Bathe et al. [8] conducted laser precision drilling experiments on TBC-coated nickel-based high-temperature alloys. They studied the relationship between hole shape, taper, and geometric features with pulse energy, pulse width, and laser repetition rate. The results demonstrated a linear relationship between hole diameter and laser power density when the pulse width was 2.0 ms. The use of high pulse energy and short pulse width lasers resulted in crack-free recast layers. Sezer et al. [9,10] investigated the influence of laser angle on the thermally affected zone, recast layer, and oxide layer characteristics in TBC-coated nickel-based high-temperature alloys. The results showed that the size of the thermally affected zone, recast layer, and oxide layer increased as the laser angle decreased. They also studied the TBC delamination mechanism.

Liu et al. [11] explored the impact of laser pulse parameters, threshold, and wavelength on the hole size using picosecond ultrashort pulse laser drilling. Their findings indicated that a laser with a wavelength of 1064 nm was required to drill holes larger than 100 μm in diameter. At a laser irradiation intensity of 6.06 J/cm^2^, the processing time was only 100 s, which was four times more efficient than at an intensity of 4.51 J/cm^2^. Moreover, under higher laser irradiation density, no recast layer was observed in blind holes and through holes, and tapering was improved. Yu et al. [12] employed a femtosecond laser with a frequency of 50 kHz to process DD6 single-crystal high-temperature alloys with TBCs. They calculated the ablation thresholds for both DD6 single-crystal high-temperature alloys and TBCs, which were found to be close (1.16 J/cm^2^ for DD6 single-crystal high-temperature alloys and 0.998 J/cm^2^ for TBCs). Marimuthu et al. [13,14] performed experiments on TBC-coated nickel-based alloys using millisecond pulse lasers. They investigated the processing thresholds for TBCs and nickel-based alloys and discussed the basic mechanisms of TBC removal using millisecond lasers. A comparison was made with water-assisted laser processing, demonstrating that water-assisted methods resulted in a smaller heat-affected zone within the substrate material [15]. Zhai et al. [16] utilized femtosecond lasers with a wavelength of 800 nm to perform impact drilling on TBC-coated nickel-based alloys. The results showed a good linear relationship between the number of femtosecond laser pulses and the processed dimensions. Through energy spectrum and Raman spectroscopy analysis, it was discovered that femtosecond lasers had minimal influence on the basic composition and phase structure of the coatings. The authors also established a finite element model based on material physical properties, comparing the experimental results with simulated data and finding good agreement. They obtained stress distribution contour maps for the micro-hole [17]. Gupta et al. [18] used a fiber laser to drill holes in nickel-based high-temperature alloys with TBCs. Their research revealed that holes created by a small number of high-energy pulse lasers had smaller entrance and exit diameters, smoother hole walls, and thinner recast layers (approximately 15 μm).

Zheng et al. [19] analyzed the crack formation mechanism during ultrafast laser drilling of TBC-coated nickel-based high-temperature alloys. They developed a related thermal-stress coupling model and explained the crack behavior in multi-layered material drilling. They proposed the use of low laser repetition rates or water-assisted methods to reduce the thermal effects during drilling. Fan et al. [20] investigated the influence of laser drilling margin on the characteristics of TBC delamination, splashing, and resolidification cracks during high-temperature alloy laser processing, combining numerical simulations with experimental methods. The results showed that the laser drilling margin effectively isolated the hole’s front edge from the ejected molten material, preventing the influence of shear stress. Das et al. [21] performed hole drilling on TBC-coated high-temperature alloys using femtosecond lasers and conducted a detailed study on the microstructural characteristics of the holes. In a 1.5 mm thick coated sample, the surface roughness of the holes created by femtosecond laser processing was less than 2 μm, with no significant accompanying damage. The presence of the holes did not affect the thermal cyclic life of the coated sample at 1100 °C. Zhang et al. [22] conducted an investigation into the ablation efficiency and quality of ablation on DD6 alloy using an ultra-short pulse laser. The study involved both experimental and theoretical analyses. The experimental results revealed a positive correlation between the ablation rate and laser energy, scanning speed, and scanning width. Conversely, the ablation efficiency demonstrated a negative correlation with laser energy. The theoretical analysis indicated that as the pulse energy increased and the scanning speed decreased, the material removal transitioned from plasma vaporization to melt removal. Additionally, the authors compared three different scanning paths and analyzed the impact of process parameters on the geometric accuracy and surface quality of the holes. The results indicated that the focal position exerted the most significant influence on the geometric accuracy and surface quality of the holes [23].

Although significant research has been conducted on the fabrication of thermal barrier coating (TBC) microholes using ultrafast lasers, there remains limited knowledge regarding the influence of laser power and scanning speed on coating damage. Thus, this study aims to investigate the impact of laser power and scanning speed on TBC micro-hole damage by employing a two-factor, four-level full factorial experimental design. The experimental plan involves utilizing picosecond lasers to create the TBC microholes, followed by a comprehensive evaluation of the hole periphery and interface condition. Specific attention will be given to the morphology of the hole periphery, as well as the extent of interface damage. The findings, in conjunction with detailed processing parameters, will shed light on the underlying patterns of damage inflicted on TBC micro-holes by varying laser power and scanning speed.

## 2. Experimental Design Translation

### 2.1. Experimental Design

This study focuses on investigating the influence of average power and scanning speed on the damage patterns of thermal barrier coating (TBC) microholes in inclined bores. A comprehensive two-factor, four-level full factorial experimental design was implemented, as illustrated in Table 1. The laser power was varied from 12 W to 30 W, while the scanning speed ranged from 80 mm/s to 320 mm/s. The substrate material employed was a high-temperature nickel-based single-crystal alloy known as DD6. The ceramic layer of the TBC consisted of yttria-stabilized zirconia (YSZ) with a nominal thickness of 150 μm, and the bonding layer material was NiCoCrAlY with a nominal thickness of 20 μm. The TBC was fabricated using a gas-phase deposition technique. The dimensions of the experimental specimens and the layout of the micro-holes are depicted in Figure 2. The depth of the machined holes was approximately 1.5 mm.

### 2.2. Experimental Methods and Experimental Equipment

The picosecond laser drilling experiment is conducted according to the aforementioned processing parameters. A commercially available diode-pumped solid-state laser from the Atlantic series (Atlantic series, Tokyo, Japan) is utilized, as depicted in Figure 3. The laser emits pulses with a wavelength of 1030 nm and a pulse width of 2.1 ps. The remaining parameters of the laser processing system can be found in Table 2. Following the completion of picosecond laser drilling, the surface morphology of the film-coated holes is examined using a coordinate measuring machine (InfiniteFocus G4, Graz, Austria). This examination includes the measurement of entrance and exit apertures and the assessment of hole surface topography. Subsequently, the samples are sectioned using wire electrical discharge machining (DK7732, Taizhou, China), as illustrated in Figure 4. Since the thermal barrier coating is a non-metallic material and lacks conductivity, it cannot be observed directly under a scanning electron microscope (SEM). Therefore, prior to SEM imaging, the samples undergo a gold-sputtering process using an ion sputter coater to enhance their conductivity, ensuring the successful preparation of the samples.

## 3. Results and Discussions

### 3.1. Effects of Process Parameters on Coating Surface Roughness

Surface roughness measurements were conducted at positions located 10 μm away from the blunt and sharp edges of the microholes, as shown in Figure 5. Figure 6 illustrates the relationship between surface roughness and process parameters on the blunt and sharp sides of the microhole circumference. The results indicate that on the blunt side, the surface roughness increases with an increment in pulse energy at a constant scanning speed. Conversely, at the same pulse energy, an increase in scanning speed leads to a reduction in surface roughness. The minimum roughness is achieved at a scanning speed of 320 mm/s and pulse energy of 12 W, which aligns with the smoother coating surface located further away from the hole edge. On the other hand, the sharp side of the microhole exhibits relatively higher surface roughness compared to the blunt side. The relationship between surface roughness and process parameters on the sharp side follows a similar trend to the blunt side but shows greater dispersion, indicating a more pronounced impact of picosecond laser processing on the surface roughness of the coating. This suggests that the coating surface on the sharp side is relatively more irregular. During the laser processing, due to the microhole structure, more ejected ablative material accumulates on the sharp side of the coating surface compared to the blunt side, resulting in higher surface roughness on the sharp side of the microhole circumference.

### 3.2. Effects of Process Parameters on Coating Microhardness

In this study, measurements were conducted on the coated surface of microholes, specifically at locations 10 μm away from the blunt and sharp edges of the holes. Two points were selected symmetrically around the microholes, with a separation of 10 μm on each side, as shown in Figure 7. A loading stress of 300 g was applied for a duration of 10 s. The average values of the two data sets were taken as the microhardness of the coating at the blunt and sharp edges, as illustrated in Figure 8. The results indicate that, overall, the microhardness of the coating at the blunt edges of the holes is generally higher compared to that at the sharp edges. Additionally, both the blunt and sharp edges exhibit similar trends, with the coating microhardness decreasing as the pulse energy and scanning speed increase. The inclined processing, influenced by the Gaussian distribution of the picosecond laser, has a more pronounced impact on the sharp edges. The molten plasma formed by laser irradiation adheres to the damaged surfaces of the acute-angle side of the hole walls and gradually accumulates, affecting the microhardness of the coating on the acute-angle side. However, due to the different physical structures, there is less accumulation on the obtuse-angle side, resulting in a lesser impact on the microhardness of the coating in that region. With increasing pulse energy and decreasing scanning speed, the thermal effects induced by the laser become more pronounced, which can affect the internal structure of the coating at the hole perimeters, resulting in defects and influencing the coating’s hardness.

### 3.3. Study on Damage to Coatings of Holes

After the perforation process, the samples undergo cutting, grinding, and polishing procedures. The cross-sections of the microholes with thermal barrier coatings are observed using a scanning electron microscope (SEM), and the results are depicted in Figure 9. The figure reveals that the picosecond laser machining of thermal barrier-coated microholes does not result in any noticeable separation between the coating and the substrate. Furthermore, the substrate does not exhibit significant signs of remelting, microcracks, or heat-affected areas. However, various degrees of damage are observed in the thermal barrier coatings. Figure 9b shows that on the obtuse-angle side, there are fine cracks on the hole wall surface due to thermal stress. However, overall, the morphology of the coating on the obtuse-angle side is superior to that on the acute-angle side, possibly attributed to the contact of the plasma with the thermal barrier coating on the acute-angle side as it is expelled from the exit, resulting in damage. Energy-dispersive X-ray spectroscopy (EDS) analysis is performed on different positions along the sharp edge of the microhole entrance (Figure 9a): position 1 corresponds to the undamaged region, position 2 corresponds to the damaged region, and position 3 corresponds to the bond coat. The results, shown in Figure 10, present the elemental composition, which is further detailed in Table 3.

Based on the spectral results, it can be observed that at position 1, the main elements are C, O, Zr, and Y, indicating the presence of the YSZ thermal barrier coating at this location. The presence of C and O elements suggests partial oxidation of the thermal barrier coating. At positions 2 and 3, similar elemental compositions are observed, with a significant amount of Ni, Co, and Cr but no Y and Zr elements. This indicates that the molten plasma formed by laser irradiation adheres to the sharp angle side of the coating damage on the wall of the air film hole. Furthermore, the similarity in elemental composition between positions 2 and 3 suggests that the adhered material is formed from the molten substrate.

Figure 11a represents the macroscopic morphology of the film cooling hole cross-section, while Figure 11b,c depicts the morphologies of the coating on the blunt angle side and the sharp angle side, respectively. To quantitatively describe the impact of laser processing on the damage to the coated film cooling hole, this study assesses the coating quality on the sharp angle side based on the area of coating damage. A triangular shape is fitted to the notch on the sharp angle side, and the area of this triangle represents the extent of the damage, as shown in Figure 11c.

Due to the lower thermal influence from the laser and minimal contact with the plasma on the blunt angle side during processing, there is no significant damage or coating delamination. However, there are microcracks formed on the hole wall surface due to thermal stress, as depicted in Figure 11d–f. The evaluation criterion for the blunt angle side is the average length of cracks starting from the bond layer and extending horizontally toward the surface.

### 3.4. Effects of Process Parameters on Damage to Coating on the Sharp Angle Side of Holes

The micrograph of damage on the sharp edge of the air-hole coating under different processing parameters is shown in Figure 12. The damaged area on the sharp edge of the air hole was measured for different parameters, and the results are presented in Table 4. From Figure 12, it can be observed that the air hole with a pulse energy of 12 W and a scanning speed of 320 mm/s exhibits the best quality, with a damage area of 95.7 μm^2^. Conversely, the air hole with a pulse energy of 30 W and a scanning speed of 80 mm/s shows the poorest quality, with a large area of damage.

The data on the damage area corresponding to different scanning speeds and pulse energies are summarized in Figure 13. From Figure 13a, it can be observed that as the scanning speed increases, the damage area on the sharp edge of the air-hole coating gradually decreases. This trend is more pronounced with lower pulse energies. At low scanning speeds, the variation in the damage area on the sharp edge of the air-hole coating is not significant for different energy levels. However, when the scanning speed reaches 240 mm/s, the damage area decreases noticeably for lower pulse energies, while the damage areas for pulse energies of 18 W, 24 W, and 30 W remain almost consistent. In Figure 13b, it can be observed that, overall, the damage area on the sharp edge of the air-hole coating gradually increases with higher pulse energies. However, at a scanning speed of 80 mm/s, the damage area for an 18 W pulse energy is smaller than that for a 12 W pulse energy, suggesting a potential error introduced during the sample preparation process.

### 3.5. Effects of Process Parameters on Damage to Coating on the Obtuse Angle Side of Holes

The morphology of crack formation on the blunt edge under different processing parameters is shown in Figure 14. The crack length on the blunt edge of the gas film hole is determined by analyzing the SEM images, and the results are presented in Figure 15. It can be observed from Figure 15 that at a scanning speed of 80 mm/s, and a pulse energy of 24 W, the average crack length on the blunt edge is the longest, exhibiting a wide dispersion of crack lengths. The maximum length of the crack can reach 60 μm, indicating the poorest processing quality of the gas film hole on the blunt edge under these parameters. On the other hand, at a scanning speed of 320 mm/s and pulse energy of 30 W, the crack length on the blunt edge is the smallest, indicating the best processing quality and the least susceptibility to damage and detachment during the thermal cycling process.

From Figure 15a, it can be observed that the crack length decreases as the scanning speed increases, indicating that a higher scanning speed leads to a better quality of the blunt edge of the gas film hole under the same pulse energy. This trend is particularly pronounced at a pulse energy of 24 W. According to Figure 15b, at scanning speeds of 80, 160, and 240 mm/s, the crack length is greatest for a pulse energy of 24 W, while at 320 mm/s, the crack length is greatest for a pulse energy of 12 W. Overall, the relationship between the crack length and laser pulse energy varies at different scanning speeds. It is speculated that this variation is a result of the combined influence of scanning speed and pulse energy on the thermal stress generated by the laser. The change in scanning speed can also affect the generation of thermal stress, thereby impacting the crack length in the coating.

## 4. Conclusions

In response to the potential coating-substrate interface defects caused by the cooling holes in thermal barrier coatings during ultrafast laser processing, the main conclusions obtained from the experimental investigation of controlling laser process parameters and evaluating the hole perimeter morphology and interface quality are as follows:(1)The roughness is the lowest when the scanning speed is 320 mm/s, and the pulse energy is 12 W. The surface roughness of the coating around the holes increases with an increase in pulse energy at the same scanning speed. The sharp edge is significantly affected by picosecond laser processing, resulting in a relatively uneven coating surface.(2)The microhardness of the coating on the obtuse-angle side of the hole perimeter is generally higher than that on the acute-angle side. This is attributed to the greater influence of molten material ejected from the substrate and accumulated on the surface of the thermal barrier coating on the acute-angle side. Furthermore, the trend of change in microhardness is the same for both blunt and sharp edges, decreasing with increasing pulse energy and scanning speed.(3)During the fabrication of thermal barrier coating film holes using a picosecond laser, due to the different ablation thresholds between the coating and substrate, the coating on the sharp edge suffers from damage and detachment due to excessive thermal stress, while microcracks are generated on the blunt edge.(4)The best quality of the hole perimeter is achieved with a pulse energy of 12 W and a scanning speed of 320 mm/s, while the worst quality is observed with a pulse energy of 30 W and a scanning speed of 80 mm/s. When using low pulse energy and high scanning speed, the damage area on the sharp edge of the gas film hole is smaller, and the crack length on the blunt edge is relatively smaller.(5)The minimum average crack length and the best processing quality on the blunt edge are achieved with a scanning speed of 320 mm/s and pulse energy of 30 W. The maximum crack length and the widest crack dispersion, indicating the poorest processing quality on the blunt edge, are observed with a scanning speed of 80 mm/s and a pulse energy of 24 W.

As an ideal method for fabricating air film cooling holes with thermal barrier coatings, ultrafast laser machining technology is significantly influenced by the selection of process parameters, which ultimately determine the machining outcome. By controlling laser process parameters in machining experiments and observing and measuring the morphology and quality of the holes, establishing the relationship between laser process parameters and the morphology and interface quality of the holes can provide reliable experimental evidence for future engineering applications of ultrafast laser machining for thermal barrier coating air film cooling holes.

## Figures and Tables

**Figure 1 materials-16-05395-f001:**
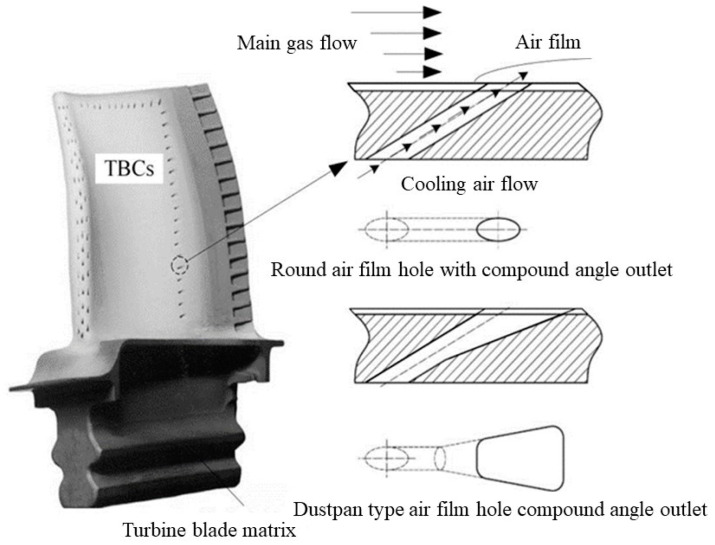
Schematic illustration of turbine blade cooling.

**Figure 2 materials-16-05395-f002:**
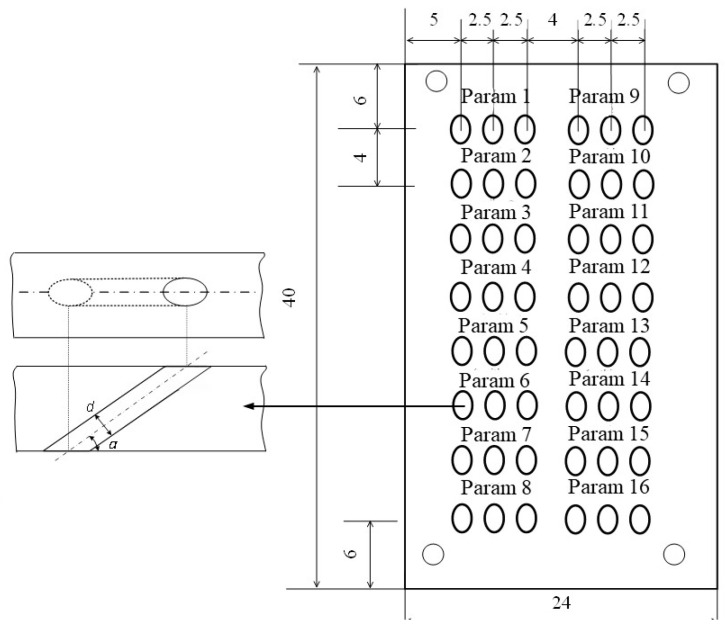
The dimensions of the coated inclined bore samples for picosecond laser processing and the arrangement of the microholes within the thermal barrier coating (TBC).

**Figure 3 materials-16-05395-f003:**
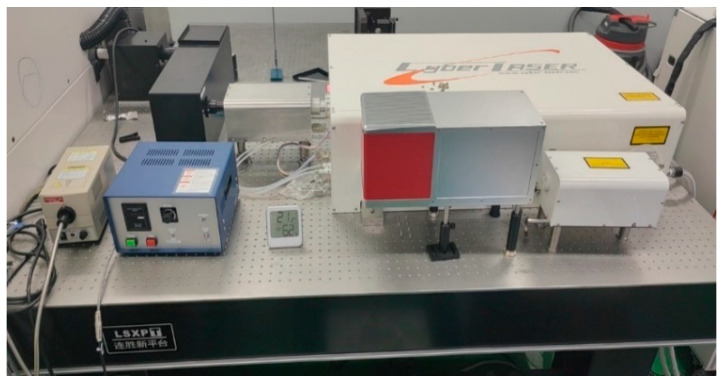
The picosecond laser emission system.

**Figure 4 materials-16-05395-f004:**
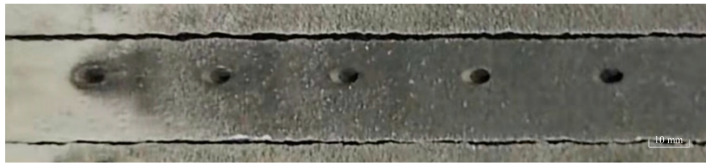
Sample after wire EDM.

**Figure 5 materials-16-05395-f005:**
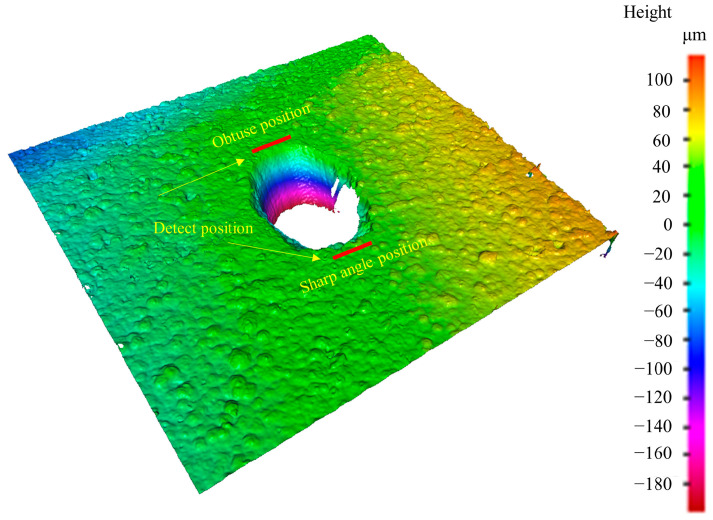
Measurement of surface roughness of microholes.

**Figure 6 materials-16-05395-f006:**
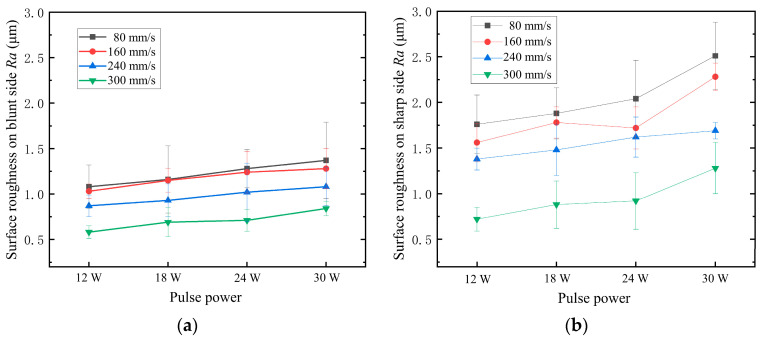
Effects of laser process parameters on surface roughness of coatings. (**a**) Obtuse position; (**b**) sharp angle position.

**Figure 7 materials-16-05395-f007:**
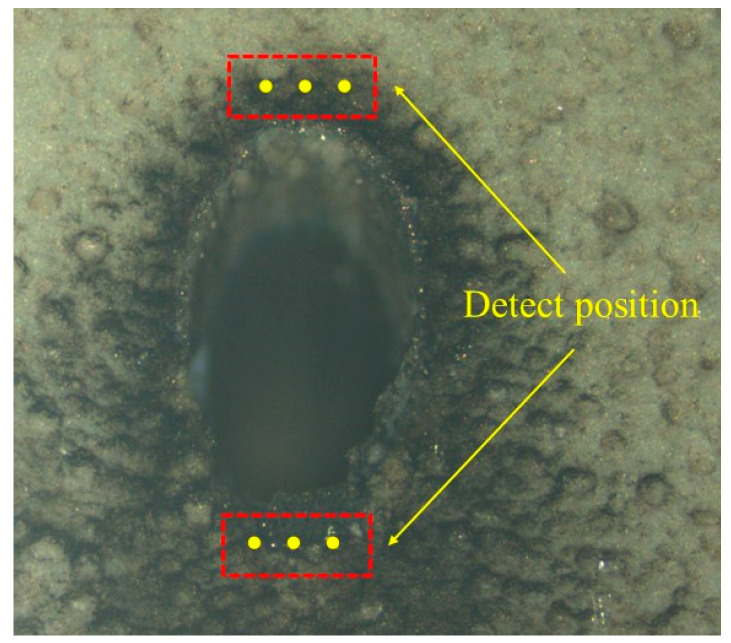
Measurement of surface microhardness of microholes (The yellow dots indicate the detect position).

**Figure 8 materials-16-05395-f008:**
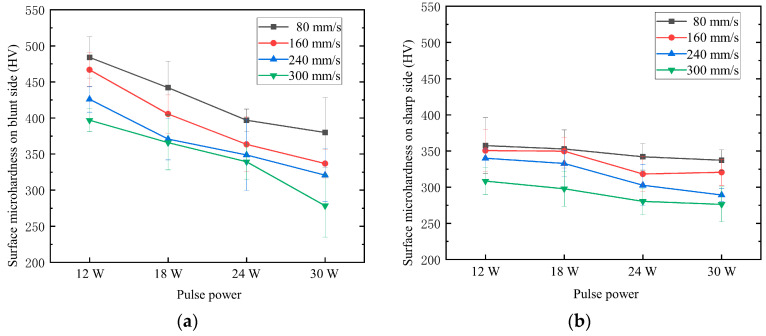
Effects of laser process parameters on surface microhardness of coatings. (**a**) Obtuse position; (**b**) sharp angle position.

**Figure 9 materials-16-05395-f009:**
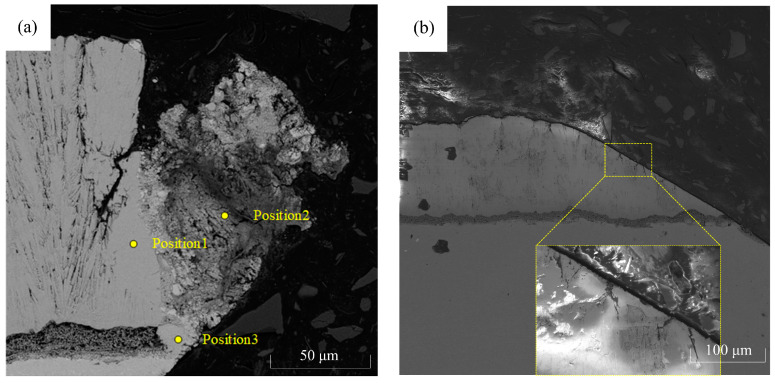
Cross-sectional images of the microholes with a thermal barrier coating. (**a**) Sharp angle side morphology; (**b**) obtuse angle side morphology.

**Figure 10 materials-16-05395-f010:**
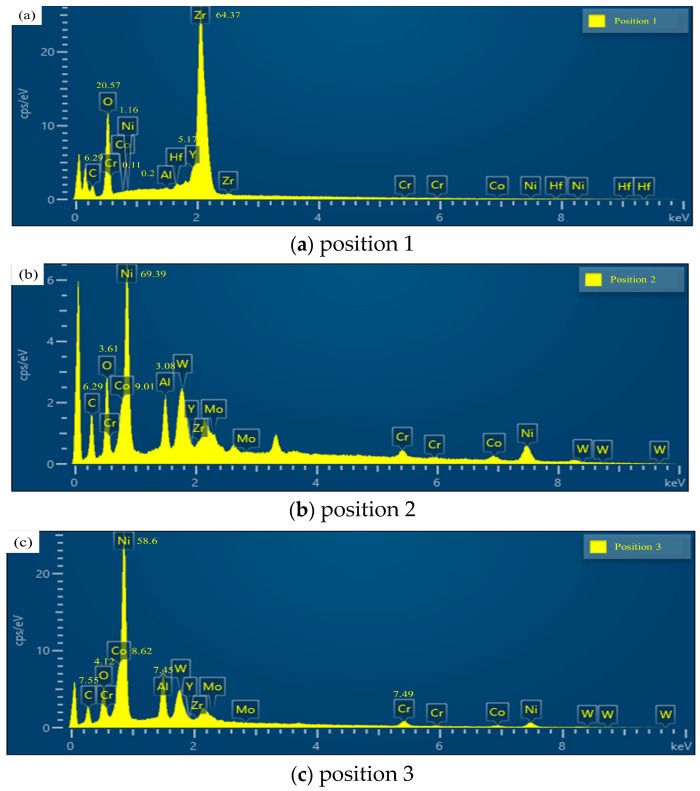
Elemental composition on the sharp angle side, with (**a**–**c**) representing the spectrum maps at positions 1, 2, and 3, respectively.

**Figure 11 materials-16-05395-f011:**
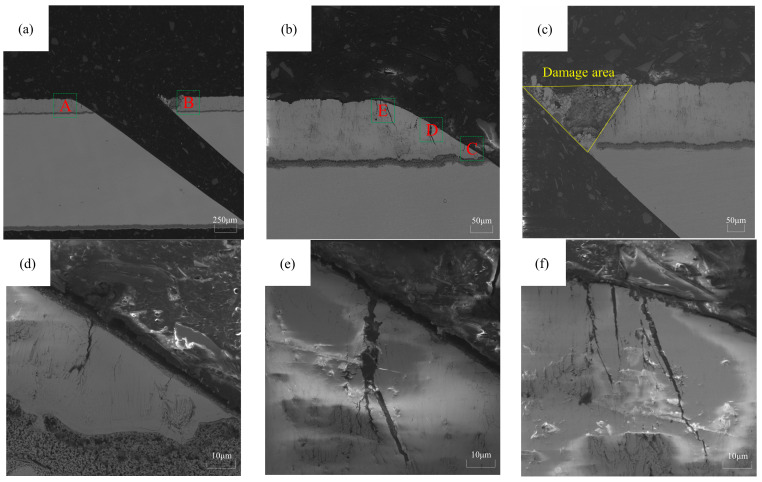
Assessment of damage to the thermal barrier coating on the microhole. (**a**) The macroscopic morphology of the cross-section of the microhole. A is the obtuse side, and B is the sharp angle site; (**b**) the morphology of the obtuse position; (**c**) the morphology of the sharp angle position; (**d**) the magnified image of region C; (**e**) the magnified image of region D; (**f**) the magnified image of region E.

**Figure 12 materials-16-05395-f012:**
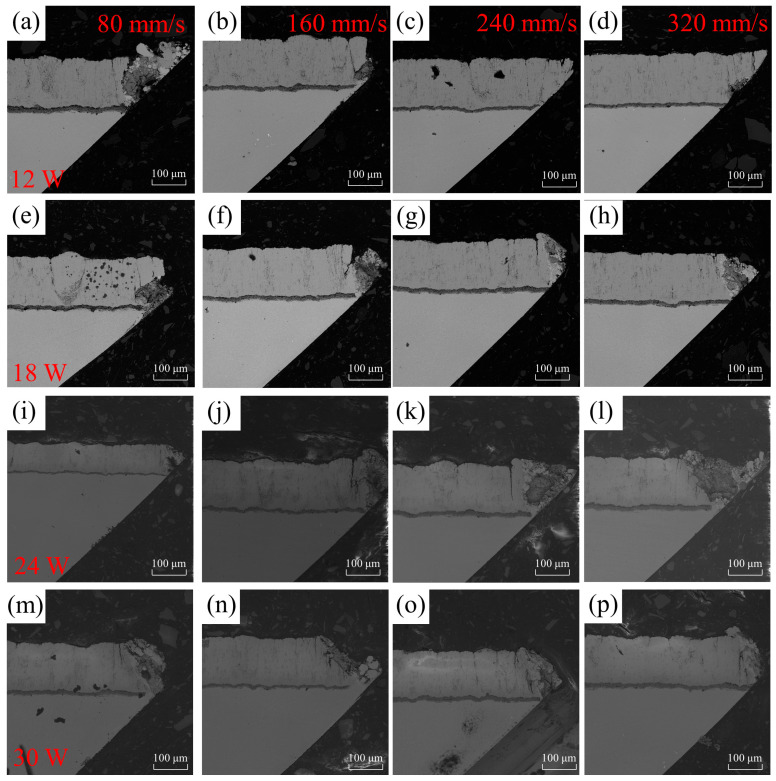
The different laser processes were evaluated for their impact on the damage morphology on the sharp angle side of the microhole. The horizontal axis represents different scanning speeds: 80, 160, 240, and 320 mm/s, while the vertical axis represents laser powers of 12 W, 18 W, 24 W, and 30 W. Subfigure (**a**–**p**) show the cross sections of the acute angle side scanning electron microscope at different scanning speeds and laser powers.

**Figure 13 materials-16-05395-f013:**
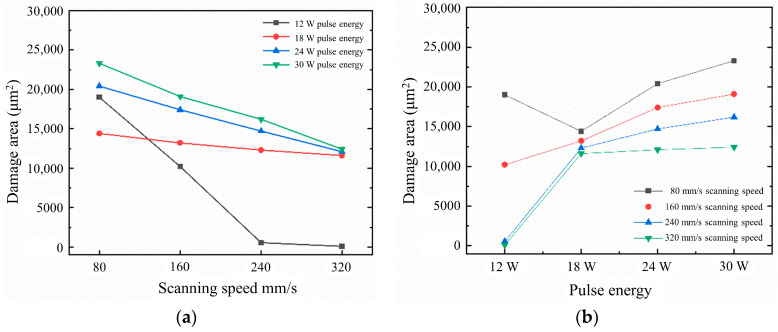
Effects of process parameters on the damage area of sharp-angle microhole. (**a**) The relationship between scanning speed and damage area; (**b**) the relationship between pulse energy and damage area.

**Figure 14 materials-16-05395-f014:**
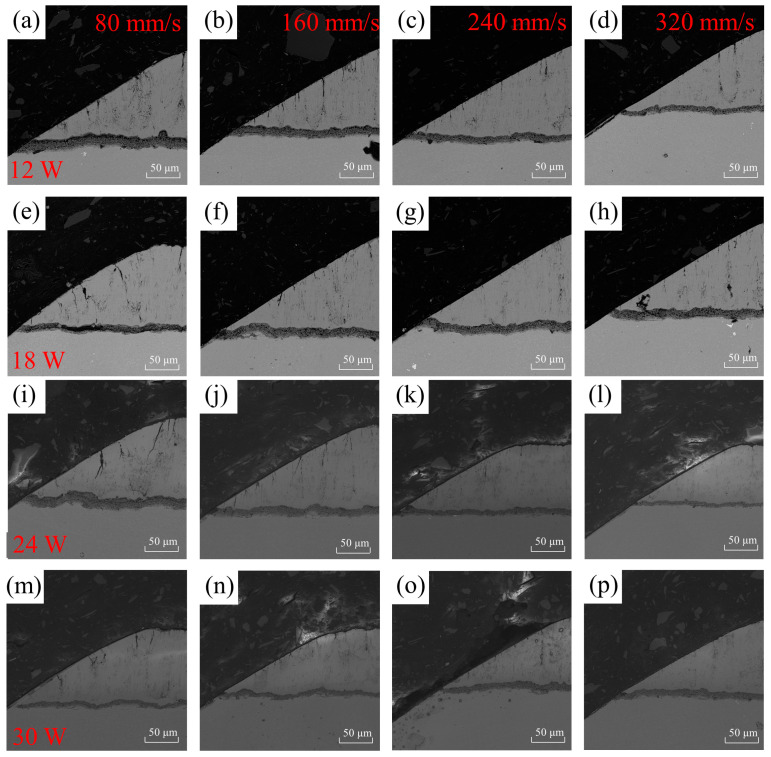
The different laser processes were evaluated for their impact on the damage morphology on the obtuse angle side of the microhole. The horizontal axis represents different scanning speeds: 80, 160, 240, and 320 mm/s, while the vertical axis represents laser powers of 12 W, 18 W, 24 W, and 30 W. Subfigure (**a**–**p**) show the cross sections of the obtuse side scanning electron microscope at different scanning speeds and laser powers.

**Figure 15 materials-16-05395-f015:**
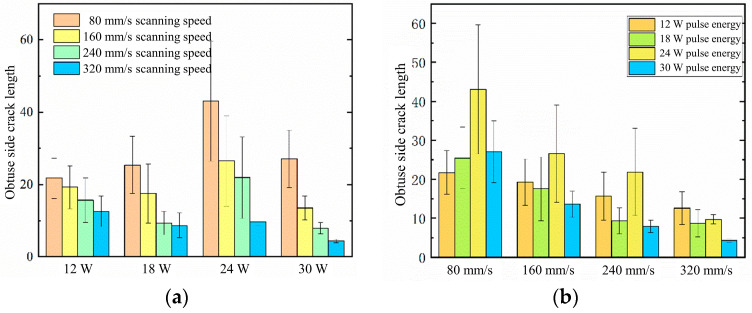
Relationship between the process parameters and damage area. (**a**) The relationship between the pulse energy and crack length on the obtuse angle side; (**b**) the relationship between the scanning speed and crack length on the obtuse angle side.

**Table 1 materials-16-05395-t001:** The experimental parameters for processing inclined bore samples with thermal barrier coating (TBC) microholes.

Parameters	Average Power	Scanning Speed
1	12 W	80 mm/s
2		160 mm/s
3		240 mm/s
4		320 mm/s
5	18 W	80 mm/s
6		160 mm/s
7		240 mm/s
8		320 mm/s
9	24 W	80 mm/s
10		160 mm/s
11		240 mm/s
12		320 mm/s
13	30 W	80 mm/s
14		160 mm/s
15		240 mm/s
16		320 mm/s

**Table 2 materials-16-05395-t002:** The basic parameters of a laser emission system.

Parameters	Symbol	Values
Pulse width	τ	2.1 ps
Diameter of the focal point	*d_f_*	48 μm
Repetition rate	*f*	75 kHz
Maximum average power	*P_max_*	30 W
Maximum peak power	$P_{max}^{pk}$	190.4 MW
Maximum pulse energy	*E_max_*	0.4 J

**Table 3 materials-16-05395-t003:** Elemental composition at different positions on the sharp angle side/Wt%.

Element	C	O	Al	Cr	Co	Ni	Y	Zr
Position 1	6.29	20.57	0.2	0.11	0	1.16	5.17	64.37
Position 2	6.29	3.61	3.08	3.62	9.01	69.39	0	0
Position 3	7.55	4.12	7.45	7.49	8.62	58.6	0	0

**Table 4 materials-16-05395-t004:** Damage area measurement on the sharp angle side.

	Scanning Speed	80 mm/s	160 mm/s	240 mm/s	320 mm/s
Pulse Energy					
40%		19,000	10,200	540	95.7
60%		14,400	13,200	12,300	11,600
80%		20,400	17,400	14,700	12,100
100%		23,300	19,100	16,200	12,400

## Data Availability

Not applicable.

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
