# Peer review of "Investigation on Thermal Barrier Coating Damages for Ultrafast Laser Drilling of Cooling Holes"

_materials, 2023, doi:10.3390/ma16155395_

Round 1

Reviewer 1 Report

The paper presents a novel technique of drilling, which is really complex to perform. The authors have still managed the same and charaterized the same.

1. How drilling was performed in the coatings? 
2. How the coating thickness was saved during drilling?
3. What is the microhardness of your coatings and what is the depth of drilled hole?

4 . Does this make impact on the bond coat also?

5. What is the porosity level and how this process is different from segmenation? Compare these processes with proper reasoning and citations?

6. Club all 3 EDS in a proper way and also club the weight % age of elements in a single image

Overall paper is good as per the standards of journal

It is already fine. NA

Author Response

Dear reviewer:

  Thanks!

Reviewer 2 Report

The authors investigated the influence of laser power and scanning speed on thermal barrier coating damage. Surface morphology, surface roughness, chemistry composition at the interface, microhardness, and damage/crack measurement have been conducted to characterize and evaluate the laser drilling process. Optimized parameters have been provided within the tested parameter region. The obtained results are interesting, and this manuscript is well-organized overall. However, there are some key discussions missing in the manuscript. To make this manuscript acceptable for Materials, the authors are required to address the following comments well.

1, Please add a SPACE between value and unit throughout the whole manuscript.

2, Please add scale bars in Figure 4 and 12.

3, The color bar in Figure 5 is too small. Please enlarge it and make it easier to read.

4, Please change the laser power from “%” to “W” in Figure 6, 14, and 15.

5, In Page 7, the discussion of microhardness is not sufficient. Please add a discussion on why coating microhardness is decreasing as pulse energy and scanning speed increase and why the blunt side is harder than the sharp side.

6, Please improve the image quality, including the figure index/scale bar alignment, unit, and font format/size, for Figure 9, 11, and 15.

7, The description of Figure 9(b) is missed.

8, In Page 9, does the material at position 2 and 3 come from the alloy part or the bonding layer part?

9, In Page 10, there is a typo in the sentence (“Due to the lesser thermal influence from the laser and minimal contact with the plasma on the blunt angle side during processing”). Please use “less” instead of “lesser”.

10, The novelty and significance of this work should be emphasized in the conclusion part.

Based on the abovementioned comments, this manuscript is recommended for major revision. A revised manuscript is required.

Author Response

Dear reviewer:

  Thanks!

Round 2

Reviewer 2 Report

The authors have answered the comments well. Now there is one more comment:

1, Please add a SPACE between value and unit for all values shown in the figures (6, 8, 12, 13, 14, 15), including the figure legend and scale bar. 

Author Response

Dear reviewer:

  Thank you very much!
